# Post-Mating Responses in Insects Induced by Seminal Fluid Proteins and Octopamine

**DOI:** 10.3390/biology12101283

**Published:** 2023-09-26

**Authors:** Guang-Xiang Guan, Xiao-Ping Yu, Dan-Ting Li

**Affiliations:** Zhejiang Provincial Key Laboratory of Biometrology and Inspection and Quarantine, College of Life Science, China Jiliang University, Hangzhou 310018, China

**Keywords:** post-mating responses, seminal fluid proteins, neurotransmitter, differential gene expression

## Abstract

**Simple Summary:**

During insect mating, male insects trigger post-mating responses in females through seminal fluid proteins, inducing a physiological transition in females that is conducive to reproduction. Simultaneously, this process indirectly increases the male’s chances of producing offspring through competition. Females, on the other hand, elicit short- and long-term post-mating responses through proteins and neural networks. Within the intricate regulatory network, seminal fluid proteins and octopamine play pivotal roles. The network spans from the female’s reproductive organs and abdominal ganglion to the brain and operates by activating receptors for seminal fluid proteins and octopamine. This activation leads to the modulation of various signals, influencing the female’s physiological behavior, organ structure, and hormone secretion, and facilitating enhanced nutrient intake, increased expression of reproductive-related genes, ovulation, and sperm storage. Ultimately, these processes aid in the successful oviposition of females.

**Abstract:**

Following insect mating, females often exhibit a series of physiological, behavioral, and gene expression changes. These post-mating responses (PMRs) are induced by seminal fluid components other than sperm, which not only form network proteins to assist sperm localization, supplement female-specific protein requirements, and facilitate the formation of specialized functional structures, but also activate neuronal signaling pathways in insects. This review primarily discusses the roles of seminal fluid proteins (SFPs) and octopamine (OA) in various PMRs in insects. It explores the regulatory mechanisms and mediation conditions by which they trigger PMRs, along with the series of gene expression differences they induce. Insect PMRs involve a transition from protein signaling to neuronal signaling, ultimately manifested through neural regulation and gene expression. The intricate signaling network formed as a result significantly influences female behavior and organ function, contributing to both successful reproduction and the outcomes of sexual conflict.

## 1. Introduction

Insect mating plays a crucial role in their overall life cycle, as it is essential for their reproduction and evolution. Before mating begins, many insects go through a complex courtship process. The success of courtship is influenced by various factors such as sound [1,2], light [3,4], sex pheromones [5], and sometimes even involves intense competition among individuals of the same species. Male insects dedicate significant efforts to ensure the propagation of their genes, including searching for mates, outcompeting rivals, providing nutritional resources to females, and guarding them post-mating to deter other males, ensuring successful fertilization and oviposition. The ultimate goal of these behaviors is to secure the continuation of their own offspring, evolving in response to ongoing sexual conflict. Female PMRs and the male molecules triggering these responses [6] have evolved under the influence of this conflict, persisting from mating until egg-laying.

Under the activation of SFPs, females undergo a complex transition into a post-mating state. This intricate process involves the recognition and breakdown of SFPs, followed by the activation of neural pathways to regulate changes in the female organ structure and secretions [7]. This continuous process guides females from ovulation to sperm release and even influences behaviors such as feeding and sleeping [8,9,10,11], facilitating successful oviposition. Males consciously adjust the levels of SFPs in response to changing environmental conditions to cope with different forms of competition [12,13]. For instance, males experience reduced reproductive capacity when they age or engage in frequent mating [14]; to compensate for this deficit, they modulate the levels of SFPs [15], thereby influencing female feeding behavior. In some cases, males reduce SFP release and instead utilize SFPs from previous mates to minimize reproductive costs [16,17].

This review will introduce some important SFPs and OA in female insects’ post-mating responses, as well as their effects on female behavior, physiological changes, responses to external stimuli, and alterations in gene expression. It will also discuss the complex interactions between protein networks and neural pathways that exist in post-mating responses. *Drosophila melanogaster*, a commonly studied model organism, holds a prominent position in understanding insect PMRs. Although other insects have also been investigated, the findings suggest a certain level of commonality in PMRs among different species.

## 2. Post-Mating Response Induced by SFPs

Insect SFPs are produced by various male reproductive tract (RT) accessory glands [18], the ejaculatory duct [19,20], ejaculatory bulb [21], and testes [22]. During mating, SFPs are transferred along with sperm into the female’s body, where they induce many aspects of the PMR in females. Firstly, SFPs alter the female’s receptivity to mating, forming a mating plug [21] to prevent the adverse effects of repeated mating on reproduction. They also aid in sperm storage and release, acting on the oviducts to enhance reproductive efficiency [23]. SFPs can influence the female’s flight, sleep, and feeding behaviors to acquire the nutrients necessary for sustaining reproduction [8,11]. Some SFPs possess antimicrobial activity [19], assisting females in boosting their immune defenses. Identified SFPs to date encompass a variety of protein categories, such as proteases/protease inhibitors, lectins, prohormone precursors, peptides, and protective proteins (e.g., antioxidants) [7,24]. These protein classes are found in seminal fluid across taxa from arthropods to mammals [25]. Relative molecular weights of different SFPs also vary significantly, ranging from 36 amino acids in sex peptide (SP) to 200–400 amino acids in prohormone-like polypeptides [26] and large glycoproteins [27]. SFPs play a crucial role in sexual conflict competition, and it is precisely for this reason that they face significant selection pressure, requiring rapid evolution to succeed in intense reproductive competition [28,29,30].

SP, one of the most extensively studied and possibly the most important SFP, is involved in almost all PMRs in *D. melanogaster* (Figure 1) [31]. The N-terminal of SP is linked to sperm, and, during sperm storage, the C-terminal of SP undergoes cleavage at a trypsin cleavage site [32]. A portion of SP enters the female’s hemolymph circulation system, where it inhibits sex peptide sensory neurons expressing the SP receptor (SPR), primarily doublesex (dsx) [33]/fruitless (fru)/pickpocket (ppk) [34,35,36] sensory neurons. This cascade results in long-lasting PMRs [37,38,39]. The SPR receptor is a G-protein-coupled receptor that functions by modulating cAMP levels through G_αi_ or G_αo_ signaling [34,35]. SPR is expressed in both the female reproductive organs and the central nervous system [34].

### 2.1. Physiological Structures and Behavioral Changes

After mating, in female *D. melanogaster*, the SP increases the feeding behavior significantly by either binding to the subesophageal ganglion associated with taste recognition and feeding [38] or through some indirect influence. The feeding amount in mated female *D. melanogaster* increases up to 2.3 times that of virgin counterparts [8]. The length of the midgut in female *D. melanogaster* gradually increases after one hour of mating and stabilizes until the sixth day, after which it starts to decrease by the tenth day. The growth of the midgut in females post-mating depends on the nutrient supply; if the nutritional status is poor, it inhibits the effects of SP on post-mating midgut growth [40]. Increased feeding and intestinal length lead to an increase in excretion [41]. Additionally, the sleep deprivation observed in female *D. melanogaster* after mating lasts for ten days [11], coinciding with the time when SP promotes midgut growth. This suggests that female insects undergo a series of long-term physiological and behavioral responses after mating. Recent studies suggest that post-mating sleep deprivation is regulated by pC1 neurons which inhibit sleep-promoting dorsal fan-shaped body (dFB) neurons to reduce sleep time. However, when the food source is changed to a higher energy sucrose, sleep deprivation disappears [42]. Evidently, females require more nutrients to ensure normal oviposition after mating, and the increase in midgut length facilitates nutrient absorption. However, increasing midgut length in poor nutritional conditions is not the preferred choice for females. The increase in activity and reduction in sleep time in females after mating are essentially aimed at acquiring more nutrients. This complex neural regulation ensures that females have sufficient energy for egg production. The physiological changes in female insects caused by mating are significant to the extent that they can influence the lifespan of females [43]. Females lacking SP in male mating tend to have a longer lifespan [44,45].

### 2.2. The Protein Network of SFPs Regulates the Storage and Release of Sperm

At the initiation of the event when male sperm enters the female’s body, SFP has no effect on the structure of the uterus or the entry of sperm into the seminal receptacle (SR) or spermathecae [46,47,48]. However, SFP plays a crucial role in the subsequent storage and release of sperm in the SR and spermathecae. On the first day after mating in *D. melanogaster*, CG33943 is required for stimulating oviposition, but it only affects females 24 h after mating [46]. Thereafter, the long-term persistence of PMRs requires seven SFPs: CG9997, CG1652, CG1656, CG17575, CG10586 (seminase), Antares (Antr), and SP [32,46,49,50]. Knocking out CG9997, CG1652, CG1656, or CG17575 in males results in defects in sperm release at an average interval of 4 and/or 10 days, while the functionality of these sperm remains normal and does not affect the female’s hatching rate [46]. Additionally, the help of these four SFPs is required for the retention of SP in the female reproductive tract (FRT) to locate the sperm storage organ (SSO) and bind to the lumens of sperm and/or SSO [47]. Subsequently, SP separates from the sperm, another critical step for sperm release. The absence of either one of these critical steps will result in defective sperm release, with the appearance of release defects due to the failure of SP detachment occurring later than in females that did not receive SP [10]. The lack of SP also leads to reduced depletion of sperm in the SR and minor damage to the mating female’s SR. Recent studies suggest that there is overlap between SFPs and certain sperm-associated proteins in the FRT fluid. Apart from this overlap, other sperm-associated proteins also contribute to sperm storage [51].

### 2.3. SPs Influence Egg Production

When SPs are lacking, the storage and release of sperm in females are affected, leading to a slower depletion of sperm and resulting in a reduced egg production [10]. However, SP also reduces female receptivity, causing females lacking SP to have a tendency for multiple matings. This behavior can compensate for some of the negative effects caused by defective sperm storage. The interaction between these factors suggests that the oviposition rate in mated females is not actually influenced by SP [44]. However, it should be noted that the process of multiple matings in females lacking SP implies the possibility of mating with multiple different males. This significantly reduces the probability of producing offspring from a particular male and increases reproductive costs. Therefore, males mix SFPs into their seminal fluid to ensure the continuation of their own offspring, albeit at a slightly increased reproductive cost.

### 2.4. SP Induced Vitellogenesis

In addition to sperm storage, post-mating vitellogenesis is also induced by SP. In virgin females, sex peptide abdominal ganglion (SAG) neurons stimulate the secretion of AstC by allatostatin C-producing thoracic ganglion (AstC-mTh) neurons, which inhibits the synthesis of juvenile hormone (JH) in corpora allata (CA). By inhibiting SAG, SP renders AstC-mTh neurons inactive [52], thereby inducing the biosynthesis of 20-hydroxyecdysone. The expression and secretion of ecdysis triggering hormone (ETH) and its receptor depend on the activation of Inka cells by 20-hydroxyecdysone. ETH acts as an obligatory allatotropin to promote the production of JH in CA [52,53], ultimately leading to the induction of vitellogenesis. JH is involved in numerous physiological processes during the growth and development of insects [54,55]. Its regulation of female JH synthesis not only affects post-mating vitellogenesis but also reduces female receptivity [56]. JH is so crucial for insects that it may influence even more PMRs. Additionally, research suggests that 20-hydroxyecdysone is produced in male accessory glands [57]. Once 20E enters the female’s body, it localizes in the mating plug and is gradually released [58]; it also impacts the female’s receptivity and oviposition behavior [59].

### 2.5. SFPs Affect Female Receptivity

SPs ultimately influence female receptivity after mating through the pathways mentioned above, and there are other SFPs that also play a role in this process. In *Helicoverpa armigera*, when the SPR in females is damaged, long-term receptivity is restored compared to the wild type. However, intriguingly, short-term receptivity remains suppressed [45]. It is PEBII, a protein involved in the formation of mating plug ejaculatory bulb, that affects short-term receptivity [60]. Furthermore, studies have shown variations in the impact of laboratory strains versus wild strains on female receptivity [61]. It is speculated that wild strains have adapted to a more complex and intense competitive environment, resulting in a stronger influence on female post-mating receptivity through SFPs.

### 2.6. SFPs Constitute the Regulatory Network of Post-Mating Response

When SFPs from the male are transferred into the female through seminal fluid, they initially reach the FRT. From there, most of the SFPs migrate to the spermathecae and SR, while a smaller portion enters the female hemolymph. The SPs that enter the hemolymph can be cleaved at the C-terminus by trypsin, and can be further broken down into smaller fragments by other proteases. However, it is not yet clear whether trypsin is upregulated in the female hemolymph post-mating or if it is derived from male SFPs transferred during mating as a means to supplement endogenous female proteases [62].

The actions of SFPs within the female are not independent, but rather form a complex network, where they interact with each other to elicit subsequent responses. During mating, the transfer of CG1652 and CG1656 from males to females requires the presence of CG9997, while the stability of CG9997 itself depends on CG1652 and CG1656 [47]. Shortly after seminal fluid entry into the female, CG1652, CG1656, CG9997, and Antr accompany SP into the SR and bind with sperm. The transfer of CG1656 and CG1652 relies on Antr and CG17575, after which they become difficult to detect within a day [63]. SP not only interacts with sperm, but also binds to the SR, a process that involves CG1652, CG1656, CG9997, and CG17575 [47]. Another SFP, CG10586, plays a crucial role in regulating protein hydrolysis. It participates in the pro-peptide cleavage of CG11864 and is involved in the proper processing of ovulin and Acp36DE. It then enters the SR and spermathecae to participate in sperm release [49]. This system, where a single protein performs multiple functions, adds to the complexity of the protein network. Geoffrey et al. utilized evolutionary rate covariation to study *D. melanogaster* SFPs, discovering and predicting several SP-related network proteins, thus demonstrating the effectiveness of evolutionary rate covariation in studying the interactions of insect SFPs [64].

## 3. OA and Their Receptors in Activating Post-Mating Responses

Octopamine (OA) possesses a biogenic monoamine structure that acts as a neurotransmitter, neuromodulator, and neurohormone in invertebrates. OA, along with tyramine (TA), is the only known biogenic amine neurotransmitter in invertebrates, and its role in invertebrates is similar to norepinephrine in vertebrates [65]. The production pathway of OA involves tyrosine being decarboxylated into TA by ty-rosine decarboxylase; then, tyramine β-hydroxylase further hydroxylates TA to convert it into OA [66] (Figure 2A). In addition to being an intermediate in OA synthesis, TA also has independent effects and can even act as an antagonist of OA [67,68]. OA is typically found only in the nervous system, while TA exists in both the nervous system and non-nervous tissues [69]. Extensive research has shown that OA and TA play important roles in insect physiology and behavior.

OA exerts its effects by binding to receptors on target cells, and the classification system for octopamine receptors is based on structural and signaling similarities to cloned insect octopamine receptors and vertebrate adrenergic receptors [70,71]. As with SPR, OA receptors belong to the G protein-coupled receptor superfamily and can be divided into three subgroups, including α-adrenergic-like octopamine receptors (OA1), β-adrenergic-like octopamine receptors (OA2) [70], and α2-adrenergic-like octopamine receptors (OA3) [72]. In *D. melanogaster*, five OA-specific G protein-coupled receptors have been identified, including OAMB-K3, OAMB-AS, Octβ1R, Octβ2R, and Octβ3R [73,74,75]. Similarly, five OA receptor genes have been identified in the *Nilaparvata lugens*, including NlOA2B2, NlOA1, NlOA2B1, NlOA2B3, and NlOA3 [76]. Among them, NlOA2B2 shows a clear sexual dimorphic expression pattern, with higher expression levels in male adults compared to female adults, and other developmental stages. OA and TA significantly increase the cAMP levels in cells expressing NlOA2B2, with OA being eight times more potent than TA and having a lower threshold for receptor activation [77]. In addition to its effects on reproduction, OA receptors also play a positive role in the insect’s stress resistance, providing some level of resilience against high temperatures, starvation, and pesticides [78].

### 3.1. OA Regulates the Contraction of Oviduct Muscles during Ovulation

The OA signal is necessary for the release of sperm from the storage organ [79], ovulation, and deposition of eggs [69,80,81]. Octopaminergic neurons in the abdominal ganglion regulate the muscle contractions of the oviducts and uterus [82], helping to trigger ovulation [81,83,84]. In *Rhodnius prolixus*, OA reduces the amplitude of oviduct contractions in a dose-dependent manner. High concentrations of OA can almost completely eliminate bursal contractions, while TA does not affect the amplitude of contractions but can slightly lower the frequency of bursal contractions [85]. Changes in OA receptors can also lead to ovulation defects in females. Following RNA interference of NlO2B2 expression in female *N. lugens*, retained eggs were observed in the ovaries, indicating ovulation defects [77]. Additionally, the lifespan of females is significantly reduced [78]. The way OA mediates ovulation in *D. melanogaster* is through the activation of Octβ2R, which increases cAMP production, activating cAMP-dependent protein kinase A (PKA) as a downstream effector. Simultaneously, it interacts with Octβ2R and OAMB to activate Ca^2+^/calmodulin-sensitive protein kinase II (CaMKII). CaMKII acts on nitric oxide synthase (NOS) to release NO, which relaxes the oviduct muscles. PKA and CaMKII work together to trigger downstream effectors that secrete fluid for egg activation and transport [86,87,88] (Figure 2B).

### 3.2. OA Causes Receptivity Decline

A decrease in OA levels leads to a decrease in receptivity in *D. melanogaster*. Compared to wild-type females, virgin females with a mutation in Tyramine β-hydroxylase (Tβh^nM18^) exhibit increased receptivity, while heterozygous mutants show intermediate receptivity, indicating a dose-dependent effect. Additionally, the mutants show some level of oviposition impairment [89]. Similar effects have been observed in *Agrotis ipsilon* Hufnagel, where injection of OA and 5-hydroxytryptamine (5-HT) increases the sensitivity of antennal lobe neurons in mated males but remains lower than in unmated males. However, newly mated males do not exhibit a significant positive response to sex pheromones after OA or 5-HT injection [90]. Furthermore, changes in receptivity may vary among insect species. Different strains of *Callosobruchus chinensis* show significant differences in receptivity changes induced by four different monoamines (dopamine, OA, TA, and serotonin) [91].

### 3.3. OA Mediates Sperm Release

The SSO in *D. melanogaster* typically consists of a singular SR and paired spermathecae [84,92]. The entry and distribution of sperm within the SSO are regulated by the nervous system [84,92]. The neuromuscular connections at the SSO are governed by the octopaminergic-tyraminergic system, although it remains unclear if the SSO also receives input from other neurotransmitters, such as glutamate [79]. Neurologically induced muscle contractions may potentially facilitate the “pumping” of sperm by the SR or the release of secretions from associated cells [93].

Both OA and TA do not play a role in uterine morphological changes or the accumulation and storage of sperm. Females with lower levels of OA and OA/TA show defects in sperm release, indicating that this is not solely a consequence of egg retention, but rather a result of the involvement of OA and TA in the process of depleting sperm from storage [79]. It is speculated that the release of sperm necessitates sustained activation of neural signals.

### 3.4. Distribution of Other Neurotransmitters in the Reproductive Organs

In addition to OA, serotonin and dromyosuppressin (DMS) are also neuromodulators present in insects. Hefetz et al. conducted a study on their levels in the reproductive tract nerve endings of female *D. melanogaster* after mating [94]. These neurotransmitters elicit different responses at various stages after mating in different regions of the reproductive tract in *D. melanogaster*, depending on the varying levels of signaling molecules. In unmated *D. melanogaster*, OA, serotonin, and DMS are distributed throughout all regions of the reproductive tract, but their distribution is not uniform. OA is most abundant in the nerve endings of the outermost oviduct, gradually decreasing until reaching the lowest concentration in the nerve endings of the uterus. Serotonin, on the other hand, exhibits more uniform distribution, with relatively higher levels in the SR. DMS is most abundant in the lateral oviducts.

At the end of mating in *D. melanogaster*, SFPs stimulate the release of octopamine at the ovarian nerve endings and trigger its accumulation at the uterine nerve endings. Serotonin levels significantly decrease in the ovary and SR. DMS decreases in the lateral oviducts and the upper and lower common oviducts, while it increases in the ovary and uterus. During the peak period of sperm storage and initiation of ovulation in *D. melanogaster*, sperm influences the level of octopamine in the nerve endings of the lateral oviducts, leading to a decrease in the lateral oviducts, upper common oviducts, SR, and uterus. Serotonin decreases in the ovary and lateral oviducts. DMS decreases in the ovary and common oviducts, while it increases in the uterus.

## 4. Regulation of Gene Expression in Insect Post-Mating Responses

Following mating, both females and males exhibit sex-specific differences in gene expression. In males, there is no differential expression of PMR genes in the head-thorax, with only a few overlapping differentially expressed genes in the abdomen. Notably, among these genes, the expression of JH esterase stands out, as it plays a crucial role in regulating JH levels and contributes to the production of SFPs. Male *D. melanogaster* demonstrates a significantly higher number of differentially expressed genes in the abdomen compared to the head-thorax, tallying up to 2068 genes with differential expression. Conversely, females exhibit the opposite pattern, with a greater number of differentially expressed genes in the head-thorax after mating compared to the abdomen [95,96].

The differentially expressed genes in males predominantly cluster around protein synthesis, which is understandable, considering the need for replenishing depleted sperm and Acps post-mating. Females, on the other hand, rely on neuroregulation to modulate their behavior after mating. Therefore, despite accepting SFPs from males via the abdomen, the communication of signals through neural pathways triggers changes in the brain. Similar findings have been observed in previous studies examining the expression profiles of honeybee queens’ brains and ovaries after mating. Ovaries respond more rapidly to mating, demonstrating gene expression primarily associated with physiological changes, while the brain’s gene expression is more behavior-related [97].

Moreover, it has been discovered that the production of SFPs is regulated by specific genes. For instance, the Hox gene *Abd-B* in the male accessory gland is instrumental in maintaining the morphology and functionality of secondary cells. These cells secrete SFPs, which are indispensable for the sustained response following female mating [50,98]. Transcriptomic analysis comparing the influence of Acps in Mexican fruit flies on females [99], along with the identification of heat-shock factor (HSF) binding sites and Hsfisoforms regulating *Anopheles gambiae*’s Acps expression [100], further support these findings.

Research indicates that the number of differentially expressed genes in the FRT of *Aedes aegypti* gradually increases after mating [101]. Similar observations have been made in female *Cephalcia chuxiongica*, where reproductive-related genes are upregulated from 1 to 24 h after mating, while immune genes remain downregulated. After 24 h post-mating, longevity-related genes start to decline [102]. It is worth mentioning that the expression of female immune genes after mating shows species-specific, and even interspecies-specific, differences [103]. In the PMR of ant queens, elevated expression of reproductive-related vitellogenin genes has been found, while defensin genes also show an increase followed by a return to normal levels within five days [104]. Similarly, in the PMR of *Spodoptera frugiperda*, aside from the downregulation of longevity-related genes, both female and male immune genes exhibit a similar pattern of upregulation followed by downregulation over time [105].

Some studies suggest that immune response genes in female *D. melanogaster* are upregulated after mating [95,106,107], while others indicate lower expression of antimicrobial peptides genes in mated female *D. melanogaster* [108]. Fedorka et al. proposed a different explanation for the changes in post-mating immune responses in *D. melanogaster*. They suggested that the observed increase in immune gene expression in mated females might be due to a discrepancy between their actual immune capabilities and their potential immune capabilities. The increased expression of immune genes in mated female *D. melanogaster* could be attributed to self-inhibition for accepting sperm or depletion of immune reserves [107]. On the contrary, Sarah et al. found that post-mating immune defense ability in female *D. melanogaster* is independent of their genetic makeup, and, interestingly, mated females display immunosuppression in a pathogen-dependent manner [109]. These differing results suggest that the expression of immune-related genes after female mating is regulated by a complex system, not solely due to the induction and regulation by SFPs. This complexity may arise from sexual conflict, where males induce females into a post-mating state while females maintain their own survival. Alternatively, the conclusions drawn by researchers could be influenced by factors such as pathogens, experimental conditions, female nutritional levels, or physiological states, resulting in different responses when faced with pathogen-induced stress.

Regarding research on *D. melanogaster* PMR, it has been found that the number of mating events can also influence gene expression. Metabolic regulatory genes involved in the metabolism of glucose, amino acids (including dopamine precursors), carboxylic acids, and the metabolic pathway of JH tend to be downregulated after the second mating event. The expression of long-term response-related genes is insensitive to the number of matings. Genes related to cell proliferation processes and reproductive cell development show an increased expression level after the second mating, and some genes exhibit significant responses specifically to the second mating event. However, this might be a delayed response to the initial mating or an overlapping effect with the aforementioned responses [110].

## 5. Conclusions

The process of post-mating reactions in female insects essentially involves adjusting their bodies to a more suitable reproductive state, but it also comes with a trade-off of compromising their chances of survival in the natural environment. The changes induced by reproductive behavior are disadvantageous for female individuals in terms of survival [111,112], whereas males, in order to ensure the production of their own offspring, may offer “gifts” or compensate for the losses incurred by females in other forms [113,114]. Both male and female insects consider the costs of mating, seeking a balance between reproduction and survival. Female insects exhibit post-mating immune suppression, which may be related to the energy expenditure involved in egg-laying [115]. Proteins secreted by the accessory gland and ejaculatory duct in male *D. melanogaster* have been found to possess antimicrobial effects [19], thus mitigating the introduction of additional pathogens through seminal fluid. The immune response exhibited by females after mating clearly represents a complex system, and the role played by SFPs in this process remains to be further discovered.

The transformation of females from never having mated to post-mating is a complex triggering process that is not exclusively mediated by SFPs or neurotransmitters alone. Instead, it involves the collaborative action of both, with the two often leading to similar outcomes. For example, SFPs can inhibit sleep after mating, while octopamine (OA), by acting on insulin-secreting neurons, can achieve the same effect [88]. Additionally, the activation of OA neurons is not necessarily directly caused by SFPs, but may occur indirectly. Acp62F, a serine protease inhibitor, participates in protein hydrolysis cascades by regulating the processing rate of ovulin [116]. After mating, ovulin in female *D. melanogaster* induces an increase in OA neuron activity, indirectly leading to an increase in the number of axonal boutons and relaxing the muscles of the oviduct to increase ovulation rates [117]. However, ovulin only produces a short-term response in oviposition, lasting 24 h [26]. The PMRs in females can be divided into neural regulation and physiological reactions, with the latter exhibiting a significantly faster response rate compared to neural regulation. Neural regulation tends to affect long-term behaviors such as feeding and sleep in females. SFPs begin processing within the male body [49] and can immediately trigger PMRs in females once transferred. In contrast, neural regulation requires more preparatory steps before activation. The involvement of a complex system of short-term PMRs may be a necessary step, as the precise and rapid initiation of PMR in females greatly increases the success rate of reproduction; after transitioning from an unmated to a mated state, maintaining the ‘switch’ in the open position is sufficient to sustain long-term responses.

Currently, most research on PMR focuses on *D. melanogaster*, while investigations on other species are limited. It remains unclear whether the regulatory mechanisms and signaling pathways of PMR are universally applicable to various insect species. The numerous SFPs pose a challenge to constructing protein signaling networks. The functions exerted by SFPs from sources other than the male accessory glands in the signaling network, as well as the interactions among SFPs downstream, have not been fully elucidated. The functions of genes such as dsx/fru/ppk and their contributions to neural signal transmission among expressing neurons remain unclear. The functionalities mediated by the activation of downstream neural systems require further refinement. Additionally, the functions of numerous differentially expressed genes after mating need to be validated. Current research has shifted from investigating the transformation of female insects after mating to understanding the changes in gene expression and their functional significance. This includes studying the co-evolution of proteins secreted by males and receptors in the female reproductive system, as well as the interconnectedness between sexual maturity and post-mating in insects.

## Figures and Tables

**Figure 1 biology-12-01283-f001:**
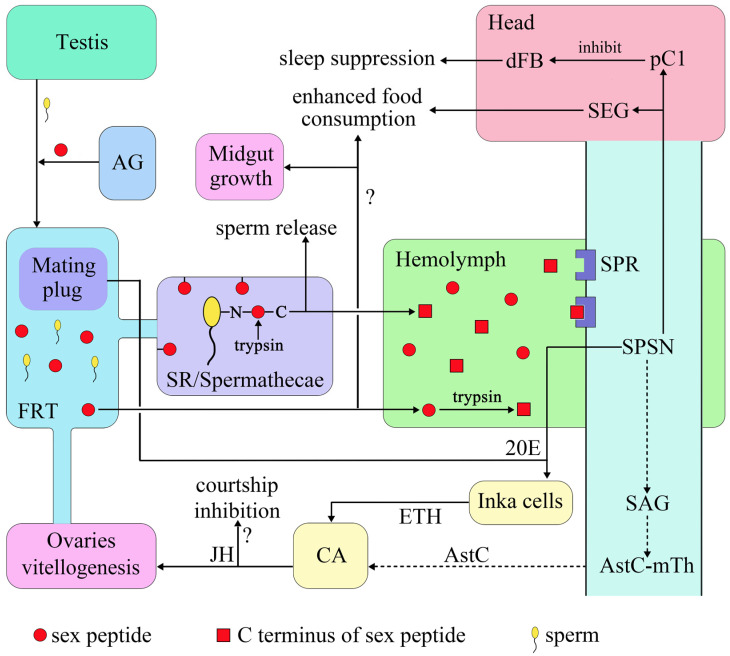
SP participate in the PMR signaling pathway of *D. melanogaster*. SP sex peptide, AG accessory gland, FRT female reproductive tract, SR seminal receptacle, SPR sex peptide receptor, SEG subesophageal ganglion, SPSN sex peptide sensory neuron, SAG sex peptide abdominal ganglion, 20E 20-hydroxyecdysone, JH juvenile hormone, CA corpora allata, ETH ecdysis triggering hormone. The symbol “?” represents an unclear pathway, the dashed line represents pathway inhibition.

**Figure 2 biology-12-01283-f002:**
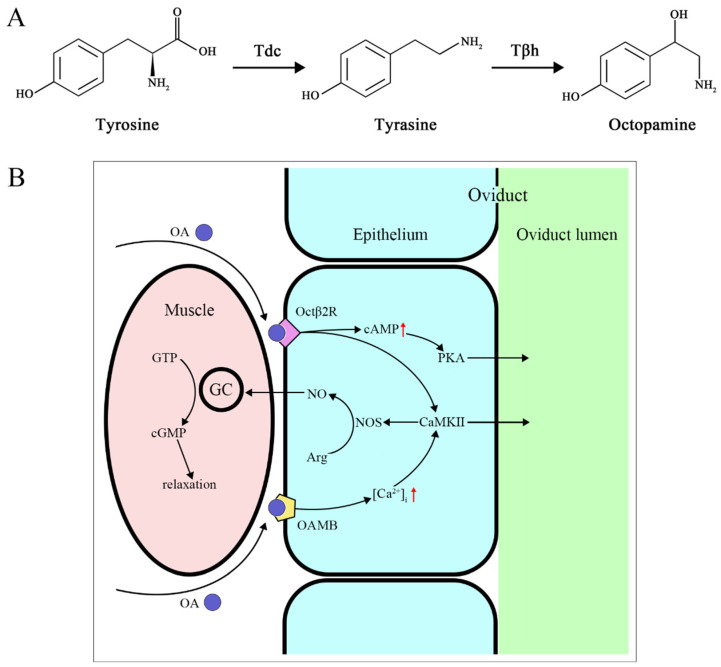
(**A**) OA biosynthetic pathway. (**B**) OA mediates oviduct relaxation through receptors. GC guanylate cyclase, NOS nitric oxide synthases. Red arrow: up-regulation.

## Data Availability

Not applicable.

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
