# Peer review of "Post-Mating Responses in Insects Induced by Seminal Fluid Proteins and Octopamine"

_biology, 2023, doi:10.3390/biology12101283_

Round 1

Reviewer 1 Report

Reviewer comment’s:

The review title address “Post-mating Responses in Insects Induced by SFPs and Neurotransmitters”.  The title of the manuscript is wider than the subject incorporated in it. Here the authors try to focus on SFPs and neurotransmitters that activate physiological responses in females following mating there by altering signaling pathways and results differential gene expression. The manuscript is well written, which I overall accept. The parameters chosen for strategy building is traditional and informative. However, I do suggest certain things which need clarification to support and strengthen the conclusion.

Abstract: Need to be improved with essential points to be highlighted. Please reframe this sentence “This review primarily focuses 23 on the responses females experience after mating, the underlying conditions and regulatory mechanisms involved, the roles of insect seminal fluid proteins (SFPs) and neurotransmitters in mediating these responses, and the resulting differential gene expression”.

Introduction: Please elaborated and update with latest updates which are missing.

Please re frame these sentences “In this review, we will introduce several important SFPs and neurotransmitters that activate physiological responses in females following mating, as well as the associated signaling pathways and resulting differential gene expression.”.

Few details on insect hormones are missing in the present text especially 20E and JH. Please incorporate it.

Please add more details on accessory gland proteins which is missing.

Please update references and details in section 2.4.

3.2. Oa Causes Receptivity Decline” please correct it to OA

Conclusion part is very weak need to the reorganized because few parts of explanations are not clear and missing latest references.

General comments:

There are many typos throughout the manuscript. In fact, the manuscript needs to be clearer to the subject chosen. The content presented here is more theatrical than expected. There are many places which needs to be updated with the latest reference. Please make the review more unique by adding more recent details. I suggest the authors to go for English corrections with a native speaker or a professional company.

The manuscript is interesting but strategic connectivity to the proposed hypothesis is lacking at certain places. Since I found some degree of difficulty in reading and understanding in certain parts of the manuscript, the article needs some corrections and certain details which needs to be incorporated. I do think that the manuscript contains important issue, interesting approaches and techniques, which can lead in understanding the role of SFPs and neurotransmitters and associated pathways. Therefore, I consider this manuscript suitable for publication after major revision in Biology.

I suggest the authors to go for English corrections with a native speaker or a professional company.

Reviewer 2 Report

The review explores the complex process of post-mating reactions (PMRs) in female insects, highlighting the trade-off between reproductive adjustments and survival. It discusses the role of seminal fluid proteins (SFPs) and neurotransmitters in triggering PMRs, with some overlapping effects. The manuscript emphasizes the need for further research in various insect species to understand the universality of these mechanisms and the intricate interactions among SFPs. It also mentions the challenges posed by the numerous SFPs and the need for validating the functions of differentially expressed genes after mating. In general manuscript is well constructed and focused on the subject. I have some minor issued marked on the pdf file. 

My major concerns are mostly related to referencing and sentence structure. I general many of the sentences are missing for references. Also authors should be careful with citing the references as many of the citations are more than 5 years old (around 80 out of 103). 

Marked on the pdf file! 

Reviewer 3 Report

This review detailly summarized responses of female flies especially in Drosophila after mating. Many critical points about post-mating responses were mentioned.  The manuscript is organized and written well. This review will benefit researchers who are interested in the field of mating-induced changes in physiology and behaviors. Some amendments are required for better presentation. I have some comments as follows.

1. lines 52-55, the sentence is not so clear, authors can correct to make it clearer. I think authors want to highlight females here?

2. line 163, Helicoverpa armigera should be italic. 

3. lines 162, SFP is not for sex peptide rather than for seminal fluid proteins.

4. lines 176, broken should be broke.

5. lines 192 and line 194, evolutionary rate covariation should be in same fromat. 

6. lines 201-202, sentence is confusing, need grammar amendment to make clearer. 

7. line 253, mating males should be mated males?

8. line 274, it is misleading, DMS is not neurotransmitter but neuropeptide. It is different. 

9. line 295, gender-specific is not so good, I think sex-specific is better.

10. lines 324-325, “…female immune genes post-mating exhibits…” is difficult to understand. 

11. line 399-340, “The roles played by proteins secreted through pathways other than the male accessory gland” the expression seems not clear.

12. line 382, “ In female D. melanogaster, post-mating” needs grammar emendation.

13. line 368 “Post-mating, female insects exhibit immune suppression” needs grammar correction.

14. line 393  “and maintaining the "switch" in…” also need grammar amendment. 

Some grammar needs correction as suggested in comments. Author can also read though the manuscript to make the manuscript better understandable.
